# Prehospital Management of Postpartum Hemorrhage—A National, Cross-Sectional Study in Norway

**DOI:** 10.3390/healthcare12181894

**Published:** 2024-09-21

**Authors:** Ann-Chatrin Linqvist Leonardsen, Laurits Dydensborg Hansen

**Affiliations:** 1Faculty of Health, Welfare and Organization, Østfold University College, P.O. Box 700, 1757 Halden, Norway; 2Østfold Hospital Trust, P.O. Box 300, 1714 Grålum, Norway; 3Innlandet Hospital Trust, Furnesvegen 26, 2380 Brumunddal, Norway; lauritsdhansen@gmail.com

**Keywords:** ambulance, external aortic compression, postpartum hemorrhage, prehospital

## Abstract

Introduction: Postpartum hemorrhage (PPH) is a critical birth complication, and is stated by the World Health Organization (WHO) as among the five most frequent causes of death during pregnancy. External aortic compression (EAC) is recommended by the WHO as an intervention to achieve temporary bleeding control. An increasing number of births outside hospital underlines the importance of competence in handling potential birth complications, such as PPH. The aim of this study was to assess prehospital personnel’s education, training, knowledge, and experiences regarding PPH and EAC across Norway. Methods: Prehospital personnel were invited to respond to a questionnaire through social media. Questions included those on education, training, knowledge, and experience regarding PPH and EAC. The Statistical Package for the Social Sciences (SPSS) version 28 was used to analyze the data, using descriptive statistics. Results: Over a two-month period, 211 prehospital personnel responded to the questionnaire, of whom 55.5% were male. The respondents had an average of 10.3 years of prehospital experience. About half of the respondents had received education (48.6%) and training (62.4%) in PPH management. Still, 95.7 percent reported a need for more education and training. On knowledge questions, only half of the responses were correct (43.7% to 60.5%). Only 21 percent of the respondents had experienced patients with PPH, and of these only 3.8 percent had used EAC. Bimanual uterine compression was the most frequent intervention used (62.5%) across hospital trusts. Conclusions: Even if prehospital personnel receive education and training in the management of PPH and EAC, almost all report needing more. The results indicate a national variation, which may be discussed as to whether it is appropriate.

## 1. Background

Postpartum hemorrhage (PPH) is a critical birth complication, and is referred to as one of the five most frequent causes of death during pregnancy [1]. One definition of PPH is bleeding >500 mL within 24 h after birth, increasing in severity up to >40 percent hemorrhage [2]. The World Health Organization (WHO) defines PPH as blood loss of ≥500 mL within the first 24 h after delivery, or any amount of hemorrhage causing hemodynamic instability. Severe PPH is defined as hemorrhage ≥1000 mL [1]. The most frequent cause is uterus atony [3]. Other causes include a retained placenta, trauma and tears in the uterus or vaginal tract, and coagulopathy [4,5]. The incidence of PPH has been reported to vary from one to five percent, depending upon the diagnostic criteria applied [6]. In Norway, the incidence of severe PPH has increased from 1.3% in 2000 to 4.9% in 2022 [5].

There are various approaches to the management of PPH, depending on the cause. These include the administration of uterotonics, uterine massage, bimanual uterine compression, insertion of an intrauterine balloon tamponade, or surgical approaches [4]. The WHO recommends the use of external aortic compression (EAC) to achieve temporary bleeding control in the management of PPH [7,8,9]. In Sweden, EAC has been implemented in national guidelines for PPH management for years [10]. However, the Norwegian Society of Gynecology and Obstetrics recommends bimanual uterine compression as the intervention of choice [2]. A 2022 systematic review [11] stated that research on EAC is limited, only identifying four studies, of which three of them focused on obstetric bleeding. These studies concluded that EAC was an effective intervention to prevent severe hemorrhagic shock and death due to PPH [12,13,14], and that EAC significantly increased the pulse rate and thereby cardiac output [15].

A study exploring prehospital birth management in Australia found an incidence of complications in 27.3 percent of cases [16]. Also, a study on paramedics indicated low mastery and a high level of anxiety regarding birth complications such as PPH [17]. Egenberg et al. [18,19,20] explored in-hospital PPH management both in low- and high-income countries, finding associations between simulation and skills training in PPH and a reduction in blood transfusions and the level of stress among healthcare professionals, as well as improved team efficacy. A study in a Norwegian county indicated that 82.8 percent of prehospital personnel (N = 87) had managed patients with PPH, and 2.9 percent of these had used EAC [21]. Moreover, results indicated that prehospital personnel lack knowledge about PPH and EAC. 

In Norway there is an increasing number of home-births and births outside hospital [5]. Also, Norway is an elongated country, with many rural locations and long travel distances from hospital. Hence, prehospital personnel require competence in the management of potential complications, such as PPH. However, there is limited research on PPH management internationally, and we have not identified any studies focusing on prehospital PPH management beyond one geographical area solely. 

### Aim

The aim of the current study was to assess prehospital personnel’s education, training, knowledge, and experiences regarding PPH and EAC across Norway.

## 2. Methods

The study had an observational, cross-sectional design, using a questionnaire. The study adheres to The Strengthening the Reporting of Observational Studies in Epidemiology (STROBE) Statement [22]. 

### 2.1. Setting and Sample

The aim was to include prehospital personnel across Norway, representing both central and rural areas. Respondents were recruited through a purposive, self-selection sampling method, through an open invitation to participate. The invitation was shared on social media through the following Facebook groups: Studentnettverk Norsk Paramedicforening (Norwegian paramedic students), #FOAMed Norway—prehospital akuttmedisin (prehospital acute medicine), Norsk Paramedicforening (the Norwegian Paramedic Association) and «Du vet du er ambulansepersonell når…» («You know you’re an ambulance personnel when…»). These groups comprise approximately 9000 followers. Inclusion criteria were as follows: unlicensed ambulance assistants, ambulance workers (vocational school), bachelor in paramedicine or nursing (180 European Credit Transfer and Accumulation System, ECTS), and paramedics (60 ECTS).

### 2.2. Data Collection

The questionnaire was based on a questionnaire developed and validated by Leonardsen et al. [21], who used it to explore the effect of simulation, e-learning, and table-tops on prehospital personnel’s competence in PPH management and EAC before and after training. Due to the slightly different approaches, the questionnaire was slightly adjusted and piloted in prehospital personnel (n = 5) with various educational backgrounds. Some of the questions relating specifically to EAC were removed, and a question about procedures used was added. (see Appendix A). Prehospital personnel in pilots found the questionnaire logical, relevant, and understandable.

Questions included those on demographics (gender, educational background, prehospital experience, and employment), questions regarding education and training/need for education and training (n = 7), knowledge questions (n = 7), and questions on experiences with PPH and EAC (n = 6). 

The questionnaire was administered through the University of Oslo’s safe digital platform «Nettskjema».

### 2.3. Analysis

The Statistical Package for the Social Sciences (SPSS) version 28 was used to analyze data. Descriptive statistics were used, and results are presented as n and percentages. Also, cross-tables were used to assess similarities or differences between hospital trusts. No methods for the calculation of missing values were used, and missing is indicated in tables as m=. 

## 3. Results

In total, 211 prehospital personnel responded to the questionnaire in the period of 10 May 2023 to 31 July 2023. Table 1 gives a description of the respondents’ demographics. 

Even though it was not possible to measure the response rate, all but one of the Norwegian hospital trusts (N = 20 out of 21 hospitals) were represented in the responses. Table 1 shows that 55.5 percent of the respondents were male, with a mean prehospital experience of 10.3 years. Most of the respondents had education from vocational school, and had full-time employment. 

### 3.1. Education and Training

Table 2 presents the results of the questions regarding education and training in PPH management and EAC. 

Table 2 shows that approximately half of the respondents had received education and/or training in PPH and EAC. However, 95.7 percent wanted more training/simulation and education. 

### 3.2. Knowledge Regarding PPH and EAC

Table 3 provides the results of the knowledge questions regarding PPH and EAC. 

Table 3 shows that from 43.7 to 60.5 percent of the respondents answered knowledge questions correctly regarding PPH and EAC, as presented by the WHO [8]. 

### 3.3. Experiences with PPH and EAC

Table 4 presents respondents’ experiences with prehospital PPH and use of EAC. 

Table 4 shows that a small number of the respondents had experienced PPH (21%), considered using EAC (9.6%), or used EAC (3.8%). The main reason for not using EAC was ‘lack of education’ (11.9%). 

### 3.4. Management across Hospitals

Table 5 provides information about prehospital interventions used across hospital trusts. 

Table 5 shows that bimanual uterine compression was the most frequent prehospital intervention included in local procedures across Norway. There were no statistically significant differences between hospital trusts regarding intervention. 

## 4. Discussion

This national study shows that even if approximately half of the respondents had received education and training in PPH and EAC, almost all of them reported needing more education and training. This was also somewhat mirrored in the knowledge questions. Very few of the respondents had experienced PPH, and even less had used EAC. Bimanual uterine compression was the intervention most prehospital services had implemented. 

Results in the current, national study show that about half of the respondents had received education (48.6%) and training (62.4%) in PPH management. Still, 95.7 percent reported a need for more education and training. In a local study by Leonardsen et al. [21], all participants had received training in PPH and EAC. Nevertheless, almost all the participants reported needing more training (97.7%) and education (96.6%). Low, self-reported competence in PPH management has been reported in previous studies as well [17,18,19,20]. Egenberg et al. [18,20] found an association between simulation and skills training in PPH and the level of stress among healthcare professionals. Moreover, Nelissen et al. [23] found an association between simulation training in PPH and a reduction in the incidence of PPH. A global consensus statement for the use of simulation-based practice in healthcare [24] emphasizes that simulation plays a crucial role in multi-agency team preparedness for the management of rare incidents. In the current study, only 21 percent had experienced PPH, even though their experience with prehospital work was a mean of 10.3 years. This indicates that simulation may be adequate to increase prehospital personnel’s preparedness for emergencies such as PPH. 

In the current study, from 43.7 to 60.5 percent of the respondents answered knowledge questions correctly regarding PPH and EAC. This may seem worrying. However, correct responses were assessed against WHO guidelines [8]. In contrast, a systematic review of guidelines on evaluation, management, and prevention of PPH shows great global variation both regarding the definition of PPH, and estimation of hemorrhage and management [25]. Also, the knowledge questions focused on EAC as an intervention. This has not been systematically implemented in prehospital services nationally. In the study by Leonardsen et al. [21], EAC was focused on due to being implemented in the local hospital based on Swedish guidelines. However, in national guidelines, EAC is suggested as an intervention in the operating room, while bimanual uterine compression is suggested during transport to the operating room [2]. This aligns with that; the latter intervention is implemented in 62.5 percent of hospital trusts, while EAC is implemented in only 21 percent. The WHO has developed evidence-based bundles for the care of PPH [26]. Here, both EAC and bimanual uterine compression are included in the «response to refractory PPH bundle». As such, the authors conclude with a need for further research to assess feasibility, acceptability, and effectiveness, as well as implementation strategies across various care interventions. 

## 5. Limitations

This study has several limitations. First, we did not perform any sample size calculations, and participants were recruited through Facebook sites. However, this method has been underlined as appropriate and useful [27]. All but one of the Norwegian hospital trusts were represented. Also, a variation in educational backgrounds was present. Still, we cannot claim the generalizability of our results. Moreover, we used a questionnaire developed to measure the effects of PPH management and EAC in a hospital where EAC had been implemented. In retrospect, the questions should not have focused on one intervention solely. 

## 6. Conclusions

This study indicates that there is a national variation in PPH evaluation, assessment, and management. However, the results are unified regarding the need for more education and training in this rare incident. This seems essential when taking into account the national increase in both home-births and PPH. 

## 7. Implications 

PPH should be included in various prehospital educational programs. Also, personnel need education and training in the evaluation, assessment, and management of PPH. Further research is needed to assess the effects of various interventions such as EAC and bimanual uterine compression, as well as the experiences of both personnel and patients. 

## Figures and Tables

**Table 1 healthcare-12-01894-t001:** Demographics of the respondents (N = 211).

Male gender, n (%)	117 (55.5)
Prehospital experience, years (mean)	10.3
Prehospital experience, years (range)	1–40
Educational background, n (%) * Ambulance assistant Ambulance personnel, vocational school Bachelor paramedicine Bachelor nursing Paramedic Other	14 (6.6) 128 (60.7) 37 (17.5) 46 (21.8) 53 (25.1) 23 (10.9)
Full-time employed, n (%)	152 (72)

* Respondents were allowed to report several educational backgrounds, for example, both vocational school and paramedic. Hence, the total number of responses to this question is higher than the number of respondents.

**Table 2 healthcare-12-01894-t002:** Responses regarding education and training in PPH and EAC (N = 210).

	n (%)
Had practical training/simulation	102 (48.6)
Had theoretical education	131 (62.4)
Has read the procedure	109 (51.9)
Had other education/training	19 (9)
Has not received training/education in PPH or EAC	32 (15.2)
Want more training/simulation in PPH and EAC	201 (95.7)
Want more education in PPH and EAC	201 (95.7)

PPH = postpartum hemorrhage. EAC = external aortic compression.

**Table 3 healthcare-12-01894-t003:** Correct answers to the knowledge questions (N = 210).

	n (%)
How many ml indicate severe PPH?	126 (60)
How do you assess the amount of blood loss in a prehospital setting? (m = 1)	110 (52.6)
What is the initial intervention in PPH? (m = 1)	119 (56.7)
When is it inappropriate to use EAC? (m = 1)	126 (60.3)
How should EAC be performed?	127 (60.5)
How can you assess if EAC is performed correctly? (m = 2)	125 (60.1)
What should be assessed when administering medications during EAC?	90 (43.7)

PPH = postpartum hemorrhage. EAC = external aortic compression.

**Table 4 healthcare-12-01894-t004:** Experiences with PPH and EAC (N = 210).

	n (%)
Experienced prehospital PPH	44 (21)
Used EAC	8 (3.8)
Considered using EAC	20 (9.6)
Had a patient where EAC could have been used	15 (7.3)
Assessed effect of EAC (N = 19) Little effect Neither/nor High effect	1 (5.2) 12 (63.2) 6 (31.6)
Reasons not to use EAC (N = 177) Lack of education Lack of training Uncertain about execution Pain or discomfort to the patient Used bimanual uterine compression Has not had any patients with PPH	21 (11.9) - 6 (3.4) 6 (3.4) 9 (5.1) 135 (76.3)

PPH = postpartum hemorrhage. EAC = external aortic compression.

**Table 5 healthcare-12-01894-t005:** Interventions used in prehospital PPH management across hospital trusts (N = 200).

	n (%)
Bimanual uterine compression	125 (62.5)
EAC	42 (21)
Other	33 (16.5)

PPH = postpartum hemorrhage. EAC = external aortic compression. No statistically significant differences between hospital trusts as measured by cross-tables, and *p*-values < 0.05 were assumed significant.

## Data Availability

Data are available upon reasonable request to the first author.

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
