# Peer review of "Prehospital Management of Postpartum Hemorrhage—A National, Cross-Sectional Study in Norway"

_healthcare, 2024, doi:10.3390/healthcare12181894_

Round 1
Reviewer 1 Report
Comments and Suggestions for Authors
Please see attached file for comments and suggestions.

The quality of the English language is sufficient.
Author Response
Reviewer 1
Comment 1
Thank you for the opportunity to review this interesting manuscript. I highly appreciate your work within this important field. I have some comments and suggestions below, that you may use to improve your manuscript.
Response:
Thank You for acknowledging our manuscript.
Comment 2
Background:
Line 38. Could it be possible to provide a newer reference than 2017? PPH is increasing in almost all countries, without anyone really knowing exactly why, so as new numbers as possible are of greatest interest. Your numbers seem very low PPH is vaginal bleeding >500 ml within 24 hours after birth. In Nor way the incidence of severe PPH increased from 37 1.7 percent in 2008 to 3.4 percent in 2017.
Response:
Thank You for this statement. We have replaced the 2017 reference with the newest available statistics from 2022.
Comment 3
Are PPH>1000 ml? For most laboring women, a blood loss of 500 ml does not pose a clinical problem, so focusing on bleedings >1000 ml seem more appropriate to me.
Response:
We agree, and the reported number in comment 2 is related to “severe PPH”, as defined by the World Health Organization (WHO) as bleeding >1000 ml. Hence, we have added the WHO definition. Please see manuscript with track changes.
Comment 4
Also, the definition of PPH is not limited to vaginal bleeding, as you state in line 33, as bleeding in cesarean births is also PPH.
Response:
Thank You. “Vaginal” has been removed.
Comment 5
Uterine massage, bimanual uterine compression, insertion of intrauterine balloon tamponade, administration of uterotonics, or surgical approaches you please put uterotonics first? It seems very illogical that it is fourth, as it is the mainstay of both prevention and treatment of PPH, as stated in your reference 3, the FIGO recommendations.
Response:
We agree and have rearranged these interventions. Please see manuscript with track changes.
Comment 6
I realise that you have affiliations with ExAC, and I much approve of Grethe Heitmans work too. However, your statement in line 46- Due to the need of applying external pressure to the uterine fundus with one hand, and internal lower uterine pressure with the other hand to compress the uterus, it may seem that EAC is less invasive to the patient. Speculations are better put in the discussion section.
Response:
We thank You for this input, and have removed this statement from the text. We do not have data supporting this statement.
Comment 7
Stated that research on EAC is limited, only identifying four studies. External aortic compression in noncompressible truncal hemorrhage and traumatic cardiac 242 arrest: A scoping review ferent focus than PPH. And, the review identify 27 studies to include in their syntheses, with three of them being obstetric studies these studies are covered by your references 12-14 (Soltan s studies) and 15. The scoping review For some trauma and obstetric situations, the use of external aortic compression in extreme circumstances is probably warranted now as a last-ditch measure pending further developments concluded that EAC was an effective intervention to prevent severe hemorrhagic shock and deaths due to PPH Please clarify this section a bit. Perhaps it is better to leave out the scoping review and just conclude based on the studies that investigate PPH.
Response:
Thank You for this comment. We have chose to keep the reference to the review, to show that we have identified relevant literature. However, we have added A 2022 systematic review (11) stated that research on EAC is limited, only identifying four studies, “of which three of them focused on obstetric bleeding”.
Comment 8
In Norway there is an increasing amount of home-births and births outside hospital
Small clinics with no operation facilities, perhaps?
Response:
Of course, this could be discussed. However, the focus in this paper is the prehospital management, not the institutional. We have not done any revisions on this issue.
Comment 9
However, there is limited research on PPH management internationally prehospital PPH. Line 68, you have not identified any studies focusing on pre-hospital PPH management nationally your reference 21: Prehospital assessment and management of postpartum
hemorrhage - , by Leonardsen et al in BMC Emergency Medicine 2021? all of Norway, as the Leonard study only covers one province?
Response:
This is correct. We have tried to clarify this in the revised manuscript. Please see manuscript with track changes.
Comment 10
Methods:
Could you please elaborate a bit more on your questionnaire? parts, both the validated questionnaire in PPH self-efficacy and the custom-made part? If so, was it exactly like Leonards? Otherwise I suggest making a translated copy of the questionnaire available as supplement material.
Response:
Thank You for underlining the unclarity. We have tried to clarify in the revised manuscript. We did not use exactly the same questionnaire. Hence, we have attached the questionnaire as supplement material, and also elaborated some more in the text. Please see manuscript with track changes.
Comment 11
Results:
The aim was to include prehospital personnel across Norway, representing both central and rural areas ata to back up that you managed that, wegian hospital trusts were represented in the responses many were from rural versus central areas, or how many from the four biggest cities versus the rest, or something like that? How many hospital trusts are there? Some info on the distribution would fit well in your Table 1.
Response:
Thank You again for declaring what’s missing to the reader. We have added information that this represents 20 of 21 hospital trusts in Norway.
Comment 12
…mean that 21% uses EAC when in table 4 only 3.8% had used EAC? Does it mean that EAC etc is described in the local guidelines or instructions? I see that the percentages add up to 100, so each respondent could only answer one thing. However, I guess it could be possible to implement both EAC and bimanual compression? Please clarify what the table shows data on.
Response:
We have added this information: “…information about interventions included in the prehospital procedures across hospital trusts”. The 21 percent corresponds to the inclusion in procedures, while the 3.8 percent corresponds to the once who have used the technique.
Comment 13
Discussion: This was somewhat also mirrored in the knowledge questions. point. Do you mean that the respondents knowledge was low? Please clarify.
Response:
Thank You once more for underlining the unclarity. We have tried to clarify in the revised manuscript. Please see manuscript with track changes.
Comment 14
In the background section, you state (line 60- A study in a Norwegian county indicated that 82.8 percent of prehospital personnel (N=87) had managed patients with PPH, and 2.9 percent of these had used EAC (21) But in your study only 21% had experienced prehospital PPH. How do you explain that?
Response:
This is a very good question, which we cannot explain. However, we have reflected upon this in the revised manuscript. Please see manuscript with track changes.
Comment 15
Limitations. You need to address the fact that openly inviting people to participate in a survey most likely introduces some bias. For example, in your sample 96% of respondents would like more education. This could suggest that mostly people with an interest in EAC has reacted to the invitation for the survey. Please comment on that.
Response:
Thank You. This has been commented in the revised limitations section. Please see manuscript with track changes.
Comment 16
Also, there are 9000 members of the Facebookgroups, where you advertised your survey. 211 participated. I guess quite a few of the Facebookmembers are members of more than one group, so it is not 9000. every prehospital unit and distribute the survey via their chief? Normally in surveys, you should aim for at least 70% participation.
Response:
This is a good point- and has been added to the limitations section. However, including through managers/chiefs is quite demanding, since the hospitals then need an assessment of the project by data protection officers, and also some places anchoring in the hospital director/head of research. Hence, this method was not chosen.
Comment 17
Lastly, I think it is important that you clearly state that it is a limitation that you asked knowledge questions with WHO as the correct answers, but also state that local guidelines vary from WHO. Does it make sense to test if local prehospital providers are familiar with the WHO guidelines for something, they rarely…You stat However, in the national guidelines, EAC is suggested as an intervention in the operating room, while bimanual uterine compression is suggested during transport to the operating room me wonder how to answer correctly in table 3? Conclusion: However, results are unified regarding the need for more education and training in this rare incident wish for more education in the probably quite selected population, you have answers from. And probably the respondents were unfamiliar with the WHO guidelines.
Response:
This is of course a timely question. However, the WHO guidelines are basis of national guidelines. We are not sure how to address this issue, other than possibly excluding questions that may vary. However, some of the questions only have one answer- which makes a basis for stating the lack of knowledge. We have added this to the limitations section.
Comment 18
Implications:
This section is well written and warranted.
Response:
Thank You.

Reviewer 2 Report
Comments and Suggestions for Authors
Dear authors,
thank you for preparing the present manuscript in which you investigate the knowledge of pre-hospital health care workers of PPH and its management.
Principally, the study is nicely prepared and conducted. Due to the design you create a certain selection bias as you may miss people without access to these platforms (maybe mention this in the limitations).
I had a few minor comments pu in the pdf. Please check and correct.

Asides of a few language pitfalls, the general use of English is good and the text is easy to understand.
Author Response
Reviewer 2
Comment 1
above you give one definition,, then you come with possible reasons. Maybe consider a smoother transsition.
Response:
This has been revised also according to Reviewer 1 input. Please see manuscript with track changes.
Comment 2
please re-phrase. I do not understand what you mean.
Response:
This has been revised also according to Reviewer 1 input. Please see manuscript with track changes.
Comment 3
I think this recommendation has to be seen in the context of world-wide emergency treatment.
Response:
Thank You for this input. We have also referred to the Swedish guidelines, which are transferable to the Norwegian context. We have added some text to meet your concerns to the limitations section. Please see manuscript with track changes.
Comment 4
please re-phrase:
maybe: lack of confidence and high level of...
Response:
Thank You for the input. We have added the suggestion.
Comment 5
please describe who you consider pre-hospital personnel, paramedics ...?
Response:
In this study, prehospital personnel correlates with personnel working in the ambulance services. We have added this to the Setting and sample section.
Comment 6
is a country with....
Response:
Revised accordingly.
Comment 7
from when to when was the sample taken?
in other words: for how long was the platform available? I see it comes in the result, but better finds a place in methods...How did you avoid double/multiple access and answering?
Response:
This information has been added to the revised manuscript. Please see manuscript with track changes.
Comment 8
What is the difference?
Maybe better write: practical training..
Response:
Revised accordingly.
Comment 9
could you support this theory by other, similar refrences from literature? I am sure in (for example) aviation there will be simulation training on problems which occur less frequent but which may be deleterious....
Response:
We thank You for this suggestion, and have added some references to transferable literature. Please see manuscript with track changes.
Comment 10
maybe connect is smoother:
...as intervention, although this...
Response:
Thank You for the suggestion. This has been revised accordingly.
Comment 11
pls mention this in the methods section.
Response:
This has been mentioned in the methods section.

Round 2
Reviewer 1 Report
Comments and Suggestions for Authors
Dear authors,
Thank you for your revision of the manuscript, and for accomodating my queries. I find that you have improved your manuscript sufficiently, and have no further comments.